# Marginal MAP Estimation for Inverse RL under Occlusion with Observer Noise

**Prasanth Sengadu Suresh**[1]                    **Prashant Doshi**[1]

[1]THINC Lab, Department of Computer Science, University of Georgia, Athens, GA 30606, USA.

## Abstract

We consider the problem of learning the behavioral preferences of an expert engaged in a task from noisy and partially-observable demonstrations. This is motivated by real-world applications such as a line robot learning from observing a human worker, where some observations are occluded by environmental elements. Furthermore, robotic perception tends to be imperfect and noisy. Previous techniques for inverse reinforcement learning (IRL) take the approach of either omitting the missing portions or inferring it as part of expectation-maximization, which tends to be slow and prone to local optima. We present a new method that generalizes the well-known Bayesian maximum-a-posteriori (MAP) IRL method by marginalizing the occluded portions of the trajectory. This is then extended with an observation model to account for perception noise. This novel application of marginal MAP (MMAP) to IRL significantly improves on the previous IRL technique under occlusion in both formative evaluations on a toy problem and in a summative evaluation on a produce sorting line task by a physical robot.

## 1 INTRODUCTION

Inverse reinforcement learning (IRL) aims to infer an expert's behavioral preferences from observations of the expert performing the task as a means to learn the expert's behavior. It represents an important paradigm in the toolkits for robot learning from demonstrations [Argall et al., 2009] and imitation learning [Hussein et al., 2017]. To perform this inference feasibly, the typical IRL methodology is to ascribe a decision-making model to the expert, which it solves optimally and follows the obtained policy [Ng et al., 2000, Abbeel and Ng, 2004]. Subsequently, IRL methods learn the reward function of the decision-making model that best explains the observed behavior under the assumption that the other components of the model are known to the learner (recent techniques relax this assumption – see Section 6.3 of the survey [Arora and Doshi, 2021] for a review of such methods).

Another assumption prevalent among IRL methods, which is particularly impractical for robot learning, is that the expert's behavior is observed fully and perfectly. Previous work [Bogert et al., 2016] has partially relaxed this assumption by recognizing that portions of the observed trajectories may be occluded from the learner's view in real-world applications. However, scarce attention has been given to the associated challenge that a robotic observer often has a noisy sensor model or a noisy perception pipeline. As such, the learner's perception of the task performance may be both incomplete and imperfect. This is different from a typical POMDP setup [Choi and Kim, 2011b] as in our case the expert can perfectly observe the environment and the noisy observations are due to the learner's imperfect sensors. For example, consider a line robot tasked with learning how to sort produce such as onions by watching the human worker perform the sort. Positioning an unobtrusive depth-camera to avoid occlusion from other nearby workers on the line is challenging and correctly discriminating between blemished and unblemished onions is not always possible.

We present the first IRL method that allows learning from trajectories, which contain both occlusions and the result of noisy perception. We adopt Bayesian IRL [Ramachandran and Amir, 2007a] as our point of departure and generalize the maximum-a-posteriori (MAP) inference [Choi and Kim, 2011a] of the reward function in two ways:

- First, we introduce a noisy observation model in the MAP inference framework under the assumption that the robot's observation noise levels are known.

- Second, given that the portions of the trajectory suffering from occlusion are known (a realistic assumption as such portions are easily detected), we may model the observed

*Accepted for the 38th Conference on Uncertainty in Artificial Intelligence* (UAI 2022).

trajectory as a full trajectory with the occluded elements marginalized. Consequently, we perform a marginalized MAP inference (MMAP) of the reward function.

A forward-backward search of the hidden variable values yields a probable list of observations that makes the MMAP inference more efficient than simply summing over all possible observations. This is done by starting from the last available observation before the occluded block, and generating a list of observations that have non-zero transition probability from it. From each observation in that list, the same is done until the end of the occluded block. This process is repeated in reverse from the available observation just after the occluded block. Only the observations common to both searches are retained.

We evaluate this MMAP-BIRL method on a toy problem and on the use-inspired domain of learning to sort produce, both modeled using discrete states and actions. We show that simply ignoring the occluded portions degrades the IRL performance — a confirmation of a similar drop in performance on a different domain is provided in [Bogert et al., 2016]. Subsequently, the MMAP-BIRL improves dramatically on a previous method that uses expectation-maximization for occlusion [Bogert et al., 2016] in both learning accuracy and run time. Our experiments with the physical cobot platform Sawyer demonstrate that the learned policy allows Sawyer to sort produce on a conveyor with both improved precision and recall.

## 2  PRELIMINARIES: BAYESIAN IRL AND MAP INFERENCE

IRL utilizes input from an expert whose behavior is assumed to be modeled using a Markov decision process (MDP) [Puterman, 1994], which it solves optimally. The expert provides demonstrations of the task to the learner and the problem is to solve for the expert's reward function that best explains the observed behavior. Formally, the MDP of an expert is defined as a quadruple $\langle S, A, T, R \rangle$, where $S$ is the set of states defining the environment, $A$ is the expert's set of possible actions, $T : S \times A \times S \to [0, 1]$ gives the transition probabilities from any given state to a next state for each action, and $R : S \times A \to \mathbb{R}$ is the reward function modeling the expert's preferences, rewards, or costs of performing an action from a state. Typically, the learner is aware of the expert's $S$, $A$, and $T$, but not $R$.

We may model the reward function as a linearly-weighted sum of $K$ basis functions [Ng et al., 2000]: $R_{\boldsymbol{\theta}}(s, a) \triangleq \sum_{k=1}^{K} \theta_k \phi_k(s, a)$ where $K$ is finite and non-zero, $\theta_k$ are the weights, and $\phi : (S, A) \to (0, 1)$ is a *feature function*. A binary feature function maps a state from the set of states $S$ and an action from the set of actions $A$ to 0 (false) or 1 (true). Notice that this representation requires pre-defining these features. An alternative is to learn

these feature functions [Levine et al., 2010] or use a neural network representation [Wulfmeier and Posner, 2015], which automatically identifies the features but typically requires far more demonstrations to converge. A (stationary) policy for a MDP is a mapping from states to actions $\pi : S \to A$ and the discounted, infinite-horizon value of a policy $\pi$ for a given reward function $R_{\boldsymbol{\theta}}$ at some state $s \in S$, with $t$ denoting time steps is given by: $E_s[V^{\pi}(s)] = E\left[\sum_{t=0}^{\infty} \gamma^t R_{\boldsymbol{\theta}}(s^t, \pi(s^t)) | \pi, s\right]$.

In this paper, we consider the situation where a portion of the trajectory is occluded from the learner. In keeping with previously established notation [Bogert et al., 2016], let the set of input trajectories of finite length $\mathcal{T}$ generated by an MDP attributed to the expert be, $\mathcal{X}^{\mathcal{T}} = \{X | X = Y \bigcup Z\}$. Here, $Y$ is the observable portion and $Z$ is the occluded part of a trajectory $X$. A complete trajectory $X$ is a sequence, $X = (s^1, a^1, s^2, a^2, s^3, ..., s^{\mathcal{T}}, a^{\mathcal{T}})$; some of these may be occluded. We build on the well-known Bayesian approach to IRL (BIRL) [Ramachandran and Amir, 2007b] that treats the reward function as a random variable and utilizes a prior distribution over the reward function, given as [1]

$$P(R) = \prod_{s \in S, a \epsilon A} Pr(R(s, a)). \quad (1)$$

Notice that the reward values for the state-action pairs are i.i.d. Ramchandran and Amir [2007b] discuss some example prior distributions including the Gaussian. We may derive the likelihood function for the demonstrated set of trajectories $\mathcal{X}$ as:

$$P(\mathcal{X}|R) = \prod_{X=1}^{|\mathcal{X}|} \prod_{t=1}^{\mathcal{T}} Pr(s_X^t, a_X^t; R)$$
$$= \prod_{X=1}^{|\mathcal{X}|} Pr(s_X^1) Pr(a_X^1 | s_X^1; R) \prod_{t=1}^{\mathcal{T}-1} Pr(s_X^{t+1} | s_X^t, a_X^t)$$
$$\times Pr(a_X^{t+1} | s_X^{t+1}; R).$$

We may rewrite this as,

$$P(\mathcal{X}|R) = \prod_{X=1}^{|\mathcal{X}|} Pr(s_X^1) \, \pi(a_X^1 | s_X^1; R)$$
$$\times \prod_{t=1}^{\mathcal{T}-1} T(s_X^t, a_X^t, s_X^{t+1}) \, \pi(a_X^{t+1} | s_X^{t+1}; R). \quad (2)$$

The policy is commonly modeled in BIRL as a Boltzmann exploration [Ramachandran and Amir, 2007b, Vroman, 2014] of the form:

$$\pi(a|s; R) = \frac{e^{\beta Q(s,a;R)}}{\sum_{a' \in A} e^{\beta Q(s,a';R)}} = \frac{e^{\beta Q(s,a;R)}}{\Xi(s)} \quad (3)$$

where $\Xi(s)$ is the partition function. As the Boltzmann temperature parameter $\beta$ becomes large, the exploration assigns increasing probability to the action(s) with the largest Q-value(s). One possible assignment to $\beta$ could be between 0 - 1 with 0 being fully exploratory and 1 being fully greedy.

---

[1] Here, BIRL's original formulation of the reward function as a function over states is extended to include both states and actions.

Methods for both maximum likelihood Vroman [2014], Jain et al. [19] and maximum-a-posteriori Choi and Kim [2011a] inferences of the reward function exist, which use the likelihood function of Eq. 2 and, in case of MAP inference, the prior as well. MAP inference for IRL has been shown to be more accurate, benefiting from its use of the prior Choi and Kim [2011a]. Formally, we may write MAP inference in log form as:

$$R^* = \arg\max_{\mathcal{R}} Pr(R|\mathcal{X})$$
$$= \arg\max_{\mathcal{R}} \ \log Pr(\mathcal{X}|R) + \log Pr(R) \qquad (4)$$

where $\mathcal{R}$ is the continuous space of reward functions, and the prior and likelihood functions are as given in Eqs. 1 and 2, respectively.

Choi et al. [2011a] presents a gradient-based approach to obtain $R^*$, which searches the reward optimality region only. Given the expert's policy, Ng and Russell [2000] show that this region can be obtained as

$$H^\pi \triangleq I - (I^A - \gamma T)(I - \gamma T^\pi)^{-1} E^\pi \qquad (5)$$

where $I$ is the identity matrix, $T$ is the transition matrix, $E^\pi$ is an $|S| \times |S||A|$ matrix with the $(s, (s', a'))$ element being 1 if s = $s'$ and $\pi(s') = a'$. $I^A$ is an $|S||A| \times |S|$ matrix constructed by stacking the $|S| \times |S|$ identity matrices $|A|$ times. The reward update rule in the gradient ascent is given as $R_{new} \leftarrow R + \delta_t \nabla_R Pr(R|\mathcal{X})$ where $\delta_t$ is an appropriate step size (or the learning rate). As computing $\nabla_R Pr(R|\mathcal{X})$ involves calculating an optimal policy, this may slow down the computations. By checking if the gradient lies within the new reward optimality region, we can reuse the same gradient and reduce the computational time: if $H^\pi \cdot R_{new} \leq 0$, then the previous gradient is reusable.

# 3 IRL UNDER OCCLUSION WITH NOISY OBSERVATIONS

Choi and Kim's MAP-BIRL assumes that the input trajectories are noiseless and fully-observable. However, these assumptions are difficult to satisfy in real-world use cases of robotic learning. In particular, it may be difficult to position a depth-camera in a factory such that the processing line task is fully observed. Furthermore, recording sensors as well as the visual state-action recognition tend to be noisy.

*Consequently, we generalize MAP-BIRL to learn in the context of both occlusions and noisy learner observations.* Let $X = (o_l^1, o_l^2, o_l^3, ..., o_l^\mathcal{T})$ where each element $o_l^t$ is the learner's observation of the expert at a time step $t$; some of these observations may be occluded. We begin by using the parameterized linear sum of reward features, $R_{\boldsymbol{\theta}}$, as the representation of the reward function. Subsequently, the prior over the reward function (Eq. 1) is now a distribution over the feature weights, decomposed into independent

distribution over each weight:

$$Pr(R_{\boldsymbol{\theta}}) = \prod_{\theta_k \in \Theta} Pr(\theta_k). \qquad (6)$$

## 3.1 FRAMEWORK

Obviously, we may simply ignore the occluded data and utilize just the observed portions of the set of trajectories $\mathcal{Y}$ for IRL [Bogert and Doshi, 2015]. In other words, $R_{\boldsymbol{\theta}}^* = \arg\max_{\boldsymbol{\theta} \in \Theta} Pr(R_{\boldsymbol{\theta}}|\mathcal{Y}) = \arg\max_{\boldsymbol{\theta} \in \Theta} Pr(\mathcal{Y}|R_{\boldsymbol{\theta}})Pr(R_{\boldsymbol{\theta}})$. But, as Bogert et al. [2016] shows, IRL's performance improves if the occluded portion can be inferred because it may contain salient state-action pairs. As such, we formulate the *marginal MAP* inference of the reward function from the data. To enable this, the likelihood of the visible portions of the trajectories can be written as the marginal of the complete trajectory $X$ by summing out the corresponding hidden portion $Z$:

$$Pr(\mathcal{Y}|R_{\boldsymbol{\theta}}) = \prod_{Y \in \mathcal{Y}} Pr(Y|R_{\boldsymbol{\theta}})$$
$$= \prod_{Y \in \mathcal{Y}} \sum_{Z \in \mathcal{Z}} Pr(Y, Z|R_{\boldsymbol{\theta}})$$
$$= \prod_{Y \in \mathcal{Y}} \sum_{Z \in \mathcal{Z}} Pr(X|R_{\boldsymbol{\theta}}).$$

Here, the parameters $\boldsymbol{\theta}$ are the maximization variables and the occluded portion $Z$ of a trajectory comprises the summation variables of the marginal MAP inference. Using the above likelihood function, the MMAP-BIRL problem is fully formulated as:

$$R_{\boldsymbol{\theta}}^* = \arg\max_{\boldsymbol{\theta} \in \Theta} \prod_{Y \in \mathcal{Y}} \sum_{Z \in \mathcal{Z}} Pr(Y, Z|R_{\boldsymbol{\theta}}) \, Pr(R_{\boldsymbol{\theta}}).$$
$$(7)$$

Let $Z$ be the collection of the observations in the occluded time steps of $X$, and $Y = X/Z$. Then,

$$R_{\boldsymbol{\theta}}^* = \arg\max_{\boldsymbol{\theta} \in \Theta} \prod_{Y \in \mathcal{Y}} \sum_{Z \in \mathcal{Z}} Pr(o_l^1, o_l^2, o_l^3, ..., o_l^\mathcal{T}|R_{\boldsymbol{\theta}})$$
$$\times Pr(R_{\boldsymbol{\theta}}).$$

The learner's observation $o_l^t$ is a noisy perception of the expert's state and action at time step $t$, and the observations are conditionally independent of each other given the expert's state and action. Therefore, we introduce the state-action pairs in the likelihood function above.

$$Pr(o_l^1, o_l^2, o_l^3, \ldots, o_l^\mathcal{T}|R_{\boldsymbol{\theta}}) = \sum_{s^1, a^1, ..., s^\mathcal{T}, a^\mathcal{T}} Pr(o_l^1, o_l^2, o_l^3,$$
$$..., o_l^\mathcal{T}, s^1, a^1, s^2, a^2, ..., s^\mathcal{T}, a^\mathcal{T}|R_{\boldsymbol{\theta}}).$$

For convenience, let $\tau$ denote the underlying (hidden) trajectory of the actual state-action pairs of the expert, $\tau =$

$(s^1, a^1, s^2, a^2, ..., s^{\mathcal{T}}, a^{\mathcal{T}})$. Then, we may reformulate the MMAP-BIRL problem as:

$$R_{\boldsymbol{\theta}}^* = \arg\max_{R_{\boldsymbol{\theta}}} \prod_{Y \in \mathcal{Y}} \sum_{Z \in \mathcal{Z}} \sum_{\tau \in (|S||A|)^{\mathcal{T}}} Pr(o_l^1, o_l^2, o_l^3,$$
$$\ldots, o_l^{\mathcal{T}}, \tau | R_{\boldsymbol{\theta}}) \; Pr(R_{\boldsymbol{\theta}}). \tag{8}$$

## 3.2 MMAP-BIRL INFERENCE

The MMAP inference problem is hard. Previous approaches, mostly in the context of Bayesian network inference, have utilized AND-OR graph structures to perform the inference [Marinescu et al., 2014]. But, the maximization variables in these techniques are discrete, which allows the use of a discrete data structure such as a graph to model the inference. As our maximization variables are continuous, we seek to solve the hard MMAP inference problem using continuous-variable optimization such as gradient ascent.

The log forms of the prior and the likelihood function in (8) are represented respectively as:

$$L_{\boldsymbol{\theta}}^{pr} = \log Pr(R_{\boldsymbol{\theta}}) \text{ and}$$
$$L_{\boldsymbol{\theta}}^{lh} = \sum_{Y \in \mathcal{Y}} \log \sum_{Z \in \mathcal{Z}} \sum_{\tau \in (|S||A|)^{\mathcal{T}}} Pr(o_l^1, o_l^2, o_l^3,$$
$$\ldots, o_l^{\mathcal{T}}, \tau | R_{\boldsymbol{\theta}}).$$

If we let the prior $Pr(\theta_k)$ in Eq. 6 be Gaussian (some values of each feature weight are more likely than others), i.e., $Pr(\theta_k; \mu_\theta, \sigma_\theta) = \frac{1}{\sqrt{2\pi}\sigma_\theta} e^{-\frac{(\theta_k - \mu_\theta)^2}{2\sigma_\theta^2}}$, where the mean $\mu_\theta$ and standard deviation $\sigma_\theta$ may differ between the feature weights. Then, the gradient of the log prior is obtained as,

$$\frac{\partial L_{\boldsymbol{\theta}}^{pr}}{\partial \theta} = \frac{-(\theta - \mu_\theta)}{2\sigma_\theta^2}. \tag{9}$$

Next, to obtain the gradient of $L_{\boldsymbol{\theta}}^{lh}$, we may expand the term $Pr(o_l^1, o_l^2, o_l^3, \ldots, o_l^{\mathcal{T}}, \tau | R_{\boldsymbol{\theta}})$ as shown below.

$$Pr(o_l^1, o_l^2, o_l^3, \ldots, o_l^{\mathcal{T}}, \tau | R_{\boldsymbol{\theta}})$$
$$= Pr(o_l^1, o_l^2, o_l^3, \ldots, o_l^{\mathcal{T}} | \tau, R_{\boldsymbol{\theta}}) \; Pr(\tau | R_{\boldsymbol{\theta}})$$
$$= Pr(o_l^1 | o_l^2, o_l^3, \ldots, o_l^{\mathcal{T}}, \tau, R_{\boldsymbol{\theta}}) \; Pr(o_l^2, o_l^3, ..., o_l^{\mathcal{T}}, \tau | R_{\boldsymbol{\theta}})$$
$$\times Pr(\tau | R_{\boldsymbol{\theta}})$$
$$= Pr(o_l^1 | s^1, a^1) \; Pr(o_l^2, o_l^3, \ldots, o_l^{\mathcal{T}}, \tau' | R_{\boldsymbol{\theta}}) \; Pr(\tau | R_{\boldsymbol{\theta}}).$$

The last step is obtained by noting that the learner's current observation is conditionally independent of its future observations given the expert's true state and action in the same time step. Let $O_l(s^1, a^1, o_l^1)$ represent $Pr(o_l^1 | s^1, a^1)$, which is the learner's stochastic mapping from the expert's state and performed action $(s^1, a^1)$ to the learner's observation of it, $o_l^1$, all corresponding to the same time step. It informs the learner about the expert's state and performed action, albeit noisily.

Using the above observation model, we may continue expanding $Pr(o_l^2, o_l^3, \ldots, o_l^{\mathcal{T}}, \tau' | R_{\boldsymbol{\theta}})$ as,

$$Pr(o_l^1, o_l^2, o^3, \ldots, o_l^{\mathcal{T}}, \tau | R_{\boldsymbol{\theta}}) = O_l(s^1, a^1, o_l^1) Pr(o_l^2, o_l^3,$$
$$\ldots, o_l^{\mathcal{T}}, \tau' | R_{\boldsymbol{\theta}}) \; Pr(\tau | R_{\boldsymbol{\theta}}) = \prod_{t=1}^{\mathcal{T}} O_l(s^t, a^t, o_l^t) \; Pr(\tau | R_{\boldsymbol{\theta}}).$$

Observe that the last term $Pr(\tau | R_{\boldsymbol{\theta}})$ corresponds exactly to $Pr(X | R_{\boldsymbol{\theta}})$ in Eq. 2 for a trajectory of state-action pairs $X \in \mathcal{X}$. Hence, we use the terms inside the outer product of Eq. 2 to substitute $Pr(\tau | R_{\boldsymbol{\theta}})$ above,

$$Pr(o_l^1, o_l^2, o_l^3, \ldots, o_l^{\mathcal{T}}, \tau | R_{\boldsymbol{\theta}}) = \prod_{t=1}^{\mathcal{T}} O_l(s^t, a^t, o_l^t) Pr(s^1)$$
$$\times \pi(a^1 | s^1; \boldsymbol{\theta}) \prod_{t'=1}^{\mathcal{T}-1} T(s^{t'}, a^{t'}, s^{t'+1}) \pi(a^{t'+1} | s^{t'+1}; \boldsymbol{\theta})$$
$$= Pr(s^1) \, \pi(a^1 | s^1; \boldsymbol{\theta}) \left( \prod_{t=1}^{\mathcal{T}-1} O_l(s^t, a^t, o_l^t) \right.$$
$$\times \; T(s^t, a^t, s^{t+1}) \, \pi(a^{t+1} | s^{t+1}; \boldsymbol{\theta}) \bigg) \; O_l(s^{\mathcal{T}}, a^{\mathcal{T}}, o_l^{\mathcal{T}}).$$

We may now rewrite the log likelihood $L_{\boldsymbol{\theta}}^{lh}$ more fully as,

$$L_{\boldsymbol{\theta}}^{lh} = \sum_{Y \in \mathcal{Y}} \log \sum_{Z \in \mathcal{Z}} \sum_{\tau \in (|S||A|)^{\mathcal{T}}} Pr(s^1) \, \pi(a^1 | s^1; \boldsymbol{\theta}) \times$$
$$\left( \prod_{t=1}^{\mathcal{T}-1} O_l(s^t, a^t, o_l^t) T(s^t, a^t, s^{t+1}) \pi(a^{t+1} | s^{t+1}; \boldsymbol{\theta}) \right)$$
$$\times O_l(s^{\mathcal{T}}, a^{\mathcal{T}}, o_l^{\mathcal{T}}). \tag{10}$$

While obtaining the gradient of $L_{\boldsymbol{\theta}}^{lh}$ is not trivial, it is possible and *we show the complete derivation of this gradient in the supplementary file.*

## 3.3 ALGORITHM FOR MMAP-BIRL USING GRADIENT ASCENT

The algorithm for MMAP-BIRL is shown in Algorithm 1. It computes the initial gradient $\nabla_{\boldsymbol{\theta}} P(R_{\boldsymbol{\theta}} | \mathcal{Y})$ for the initially sampled weights (line 1), performs forward rollout (line 7) to find the policy corresponding to these weights, computes the reward optimality region (line 8) and stores all of these (line 9). Notice that the gradient requires inferring possible $Z$, a set considerably narrowed down by forward-backward search. Then, we repeatedly update the reward weights according to the update rule (line 11). If the weights fail to satisfy the optimality condition (line 13), we find a new gradient and proceed with the remaining steps. On convergence, we return the learned weights (line 21).

We briefly analyze the complexity of this algorithm, which consists of two main components: (1) obtain the gradient and its optimality region using MMAP, and (2) reuse a

**Algorithm 1:** MMAP-BIRL

---

**Input:** MDP, $\mathcal{Y}$, step-size $\delta_n, \epsilon$
**Output:** Learned reward function $R_{\boldsymbol{\theta}}$

---

1  Sample $R_{\boldsymbol{\theta}}$ from the prior distribution
2  Initialize $\Pi \leftarrow \varnothing, \delta \leftarrow \infty$
3  **if** $\nabla_{\boldsymbol{\theta}} Pr(R_{\boldsymbol{\theta}}|\mathcal{Y})$ *not in* $\Pi$ **then**
4     **if** *no. of occlusions > 0* **then**
5        $\mathcal{Z} \leftarrow$ Bidirectional_Search(MDP, $\mathcal{Y}$)
6        $\nabla_{\boldsymbol{\theta}} Pr(R_{\boldsymbol{\theta}}|\mathcal{Y}) \leftarrow$ Compute_MMAP
         _Gradient(MDP, $\mathcal{Y}$, $\mathcal{Z}$)
7  $\pi \leftarrow$ Solve_MDP($R_{\boldsymbol{\theta}}$)
8  $H^{\pi} \leftarrow$ Compute_Reward_Optimality_Region($\pi$) as
   shown in Eq. 5
9  $\Pi \leftarrow \{\langle \pi, H^{\pi}, \nabla_{\boldsymbol{\theta}} P(R_{\boldsymbol{\theta}}|\mathcal{Y})\rangle\}$
10 **while** $\delta > \epsilon(1-\gamma)/\gamma$ **do**
11    $R_{\boldsymbol{\theta}_{new}} \leftarrow R_{\boldsymbol{\theta}} + \delta_n \nabla_{\boldsymbol{\theta}} Pr(R_{\boldsymbol{\theta}}|\mathcal{Y})$
12    **if** $R_{\boldsymbol{\theta}_{new}}$ *is not in the reward optimality region* $H^{\pi}$
      **then**
13       Repeat steps 3 to 8 using $R_{\boldsymbol{\theta},new}$
14       **if** *isNewEntry($\langle \pi, H^{\pi}, \nabla_{\boldsymbol{\theta}} P(R_{\boldsymbol{\theta}_{new}}|\mathcal{Y})\rangle$)* **then**
15          add $\langle \pi, H^{\pi}, \nabla_{\boldsymbol{\theta}} P(R_{\boldsymbol{\theta}_{new}}|\mathcal{Y})\rangle$ to $\Pi$
16    **else**
17       ReuseCachedGradient($\Pi$)
18    $\delta \leftarrow |R_{\boldsymbol{\theta}} - R_{\boldsymbol{\theta}_{new}}|$
19    $R_{\boldsymbol{\theta}} \leftarrow R_{\boldsymbol{\theta}_{new}}$
20 **return** $R_{\boldsymbol{\theta}}$

---

cached gradient if $R_{\boldsymbol{\theta}_{new}}$ is within the reward optimality region, otherwise recalculate the gradient. The first part involves a forward-backward search that obtains the list of possible candidates for the occluded block. Its complexity in the worst case is $\mathcal{O}((|O_l||S||A|)^{d/2})$ where $d$ is the length of the occluded block, but typically less because not all observations are always possible. Computing the gradient involves parsing all the features for each trajectory whose complexity is $K\mathcal{T}|\mathcal{X}|$. Next, solving the MDP using policy iteration with current reward weights has the policy space as the worst-case complexity $\mathcal{O}((|S|^{|A|})$ but more typically its $\mathcal{O}(|S| \cdot N_{iter})$, where the latter term denotes the number of iterations until convergence, which is not fixed. And, the complexity of the optimality region $H^{\pi}$ computation is dominated by the multiplication of two matrices whose complexity is $\mathcal{O}(|S|^3|A|)$. The complexity of the second part involves updating the reward weights using the previously computed gradient, whose complexity is $K$ and checking if $R_{\boldsymbol{\theta}_{new}}$ is within the range of one of the cached entries in $\Pi$, whose complexity is $\mathcal{O}(K|\Pi|)$. If a cached gradient cannot be used, then the complexity of the first part again applies. As such, the overall complexity of MMAP-BIRL is dominated by the complexity of solving the MDP and computing the optimality region scaled by how many times these operations must be performed in the gradient ascent process.

# 4 EXPERIMENTS

We evaluated MMAP-BIRL on two domains. For both these domains, we use the Boltzmann temperature $\beta = 0.3$, step size $\delta_n = 0.01$, discount factor $\gamma = 0.99$, and a decay rate of 0.95. We solve the MDP in Algorithm 1 (line 7) using policy iteration to obtain the current iteration's policy, $Q$- and $V$- values. Our code for this algorithm can be accessed at this `https://github.com/prasuchit/mmap-birl/`.

We evaluate the performance of MMAP-BIRL using the well-known metric of inverse learning error (ILE) and run time. ILE is inversely proportional to the accuracy of the learned reward function, ILE $= \sum_{s \in S} \|V^{\pi_E}(s) - V^{\pi_L}(s)\|$, where $V^{\pi_E}$ is the value of the expert's policy $\pi_E$ and $V^{\pi_L}$ is the value of the learned policy $\pi_L$ using the true MDP.

We compare MMAP-BIRL's performance with an extension of HiddenDataEM [Bogert and Doshi, 2017] that utilizes expectation-maximization for managing the hidden portions of the trajectory. However, as the method does not account for observer noise, we generalize it by introducing the observation model $O_l$ into the method. MMAP-BIRL's performance is also compared with Choi and Kim's MAP-BIRL, which serves as an ablated baseline to establish the value of modeling observation noise and occlusion for IRL.

## 4.1 FOREST WORLD

Our domain for formative evaluations is a previously introduced toy problem [Bogert et al., 2016] consisting of a 4x4 grid traversed by a fugitive. A UAV is tasked with reconnaissance of the fugitive (to learn which location is the goal and which locations are avoided), but the latter's movement is not always visible due to forest cover in some sectors, as shown in Fig. 1a. We model the fugitive's navigation through the grid as an MDP. The states of the MDP are the sector coordinates $(x, y)$ and there are four actions corresponding to movement in the 4 cardinal directions. The start state of the fugitive may vary. However, the resulting next sector location is not deterministic: there is a 10% chance that the fugitive may end up in any of the three sectors other than the intended one. A hidden tunnel from (2,3) to (3,3) introduces ambiguity whether the fugitive has reached the goal location of (3,3): with a chance of 30% the UAV's sensors may incorrectly place the fugitive back at (2,3). This forms the UAV camera's observation model. The UAV models the fugitive's reward function as a weighted linear sum of the following two feature functions:

- Avoidable_state$(x, y)$ is activated if the fugitive eschews $(x, y)$,

- Goal_state$(x, y)$ is activated if $(x, y)$ is the fugitive's goal location.

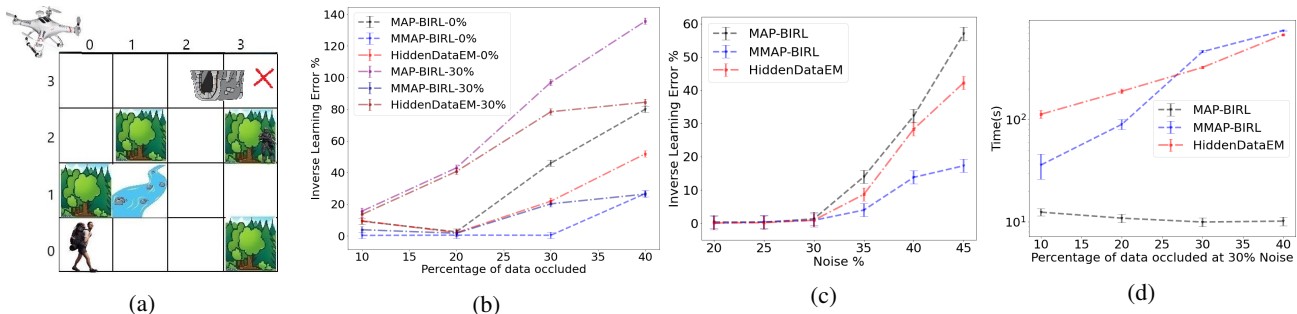

Figure 1: (a) A fugitive intends to reach the safe sector (3,3) while avoiding the river in (1,1) and the army personnel in (3,2). These sector preferences are not known to the UAV flying overhead. (b) ILE increases with increasing occlusions on noise-free data and data with 30% noise, but less so for MMAP-BIRL. (c) ILE changes with increasing noise on occlusion-free data. (d) Average clock times for increasing occlusion at 30% noise. These were measured on a Ubuntu PC with quad-core Xeon CPU @ 3.2GHz and 76GB RAM. The vertical error bars denote one standard deviation.

For Fig. 1a, the fugitive's own reward function places a high negative weight on Avoidable_state$(1,1)$ and Avoidable_state$(3,2)$, whereas a high positive weight for Goal_state$(3,3)$.

A simple comparison of the learned weights can be done as we have access to the expert's true reward weights in a toy domain. We compare along the 3 features, in a setting of 30% noise and 4 occlusions per trajectory for a total of 10 trajectories with 15 steps each. MMAP-BIRL produces learned reward weights [-0.7181, -0.8397, 0.6902] as compared to the true weights of [-0.5, -1, 0.1]. However, in order to compare them on a common scale, we compare the softmax values of both rewards. This shows a difference of 0.2131 along feature 1, 0.0065 along feature 2 and -0.2196 along feature 3. Thus the learned weights induce a slightly lower reward for the first two features and slightly higher one for the goal feature, nonetheless, maintaining the general trend of the rewards.

We evaluate the performance of the methods under varying levels of occlusion (from 10% to 40%) while keeping observation noise fixed at 0% and 30% (Fig. 1b), and for varying levels of noise (from 20% to 45%) without occlusion (Fig. 1c). Each data point is the mean of 10 batches with 10 trajectories in each batch exhibiting the corresponding level of occlusion and noise. A Gaussian prior ($\mu = -1$, $\sigma^2 = .5$) is used for MAP- and MMAP-BIRL. As we may expect, the ILE increases as the learning becomes more challenging. This increase is worst for MAP-BIRL, which does not model the noise and simply ignores the occlusion. Between the two methods that model both, MMAP-BIRL exhibits a much lower ILE, and it does not increase as dramatically as HiddenDataEM, especially for the 30%-noise level. However, the HiddenDataEM does run marginally faster than MMAP-BIRL in this toy problem, and both show run times that generally increase *linearly* as the occlusion increases. As we may expect, MAP-BIRL's run times remain mostly consistent as it does not reason about the uncertainty.

## 4.2 ROBOTIC SORTING ON PROCESSING LINES

Our second evaluation domain is a use-inspired robotic line sorting where the physical cobot Sawyer (from Rethink Robotics) is tasked with sorting onions on a conveyor belt after observing a human perform the sort. Sawyer observes the human using a Kinect v2 RGB-D camera, and a trained YOLO v5 deep neural network model [Redmon et al., 2016] is used to detect and classify the onions as blemished or not. The depth-camera frames are quantized into appropriate state variables by SA-Net [Soans et al., 2020]. These state recognitions are shown in red text on the frames in Fig. 2.

We model the onion-sorting domain as a discrete MDP as follows. The factored state is captured by 3 key variables yielding a total of 48 states: *Onion_location*: {on conveyor, hover location, in front of face, in bin}; *EndEffector_location*: {on conveyor, hover location, in front of face, in bin}; and *Prediction*:{good, bad, unknown}. The value "hover location" is a region of space just on top of the conveyor and "in bin" indicates just inside a bin that holds discarded onions. An example state where the onion is on the conveyor, the end-effector is in the hover location, and the prediction is blemished would be represented as (on conveyor,hover location,bad). Prior to inspection, every onion's condition is unknown. The sorter performs one of five abstract actions: *claim new onion*, which shifts the sorter's focus to a new onion, *Pick up the onion*, which denotes the sorter grasping the onion; *Inspect after picking* the onion by rotating it and checking for blemishes; *Place onion on the conveyor* after it is picked; and *Place onion in the bin* after it is picked. [2]

We utilize the following six Boolean feature functions to represent the human sorter's preferences:

---

[2]Our MDP does not provide low-level control, which allows its variables to remain discrete. We follow a pipeline programming architecture in which the MDP's abstract actions map to motion planners in joint angle space and plan in real time.

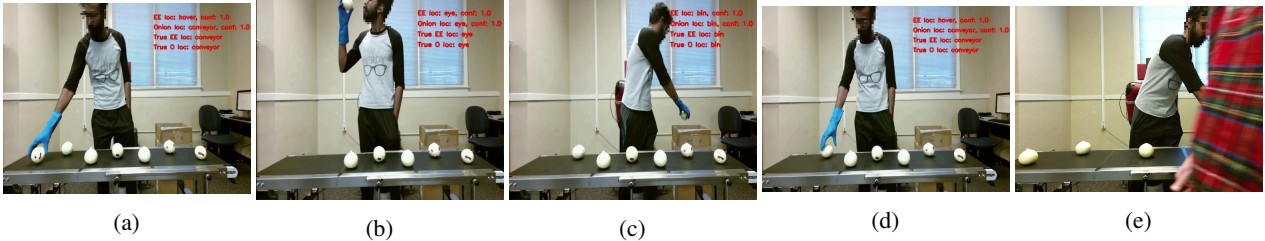

(a)      (b)      (c)      (d)      (e)

Figure 2: (a–d) These frames show a human picking an onion, inspecting it, placing it after making a decision, and choosing the next onion in sequence. The red text appearing on the images is the state predicted by SA-Net. (e) An example occluded frame where SA-Net is unable to make a prediction. At this point the expert could be placing the onion back on the conveyor or in the bin.

- Good onion placed on conveyor$(s, a)$ is 1 when a good onion is placed back on the conveyor,
- Bad onion placed on conveyor$(s, a)$ is 1 when a bad onion is placed back on the conveyor,
- Good onion placed in bin$(s, a)$ is 1 when a good onion is placed in the bin,
- Bad onion placed in bin$(s, a)$ is 1 when a bad onion is placed in the bin,
- Claim new onion$(s, a)$ is activated when a new onion is chosen if no onion is currently in focus,
- Pick if unknown$(s, a)$ is 1 if the considered onion, whose classification is unknown, is picked.

The observer noise in this domain comes from YOLO sometimes misclassifying onions due to changing lighting conditions and SA-Net incorrectly identifying the state; we estimated this noise empirically to be approximately 30%. This makes the state estimation uncertain and is recorded as an observation. This forms the probabilistic observation model of the camera. Occlusions occur when another person inadvertently passes by in front of the camera during the recording and blocks a frame either partially or fully, which leaves SA-Net unable to ascertain a state value (as shown in Fig. 2e).

We recorded 12 trajectories from human demonstrations with an average of 4 state occlusions (due to the person blocking) in 135 state-action pairs per trajectory. Figure 4a compares ILE between MMAP-BIRL, the extended HiddenDataEM, and MAP-BIRL on these trajectories. For purposes of the evaluation, greater occlusions were obtained by removing states in the trajectories at random. First, the degraded performance of MAP-BIRL due to noise and occlusion is evident from the significantly higher ILE shown by the technique. Notice that MMAP-BIRL continues to show a significantly lower ILE in comparison to HiddenDataEM on this larger domain. Equally important, it does so in much less time showing more than an order of magnitude in speed up, as is evident from Fig. 4b. Furthermore, the run times increase linearly in general for both methods as the rate of occlusion increases.

We let Sawyer physically sort through 50 faux onions using the policies learned by both MMAP-BIRL and HiddenDataEM from the 12 recorded and processed trajectories **(see the sort video in the supplementary file)**.

| Method | (TP,FP,TN,FN) | Precision | Recall |
|---|---|---|---|
| MMAP-BIRL | (23,2,18,7) | **0.92** | **0.767** |
| HiddenDataEM | (16,10,15,9) | 0.615 | 0.64 |

Table 1: Precision and recall of Sawyer physically sorting 50 onions on a conveyor using policies from rewards learned by MMAP-BIRL and HiddenDataEM, respectively. TP - True positive, FP - False positive, TN - True negative, FN - False negative.

Sawyer performs the sort by receiving the bounding boxes from YOLO, on which techniques such as central orthogonal projection, and direct linear and affine transforms are used to obtain the coordinates of the onions in its 3D workspace. The robot picks up the onions and places them either in the bin or back on the conveyor after inspection. Figure 3 shows the bounding boxes detected in real-time by YOLO [Redmon et al., 2016]. Sorting performance is measured using the domain-specific metrics of precision and recall where $precision = \frac{TP}{TP+FP}$ and $recall = \frac{TP}{TP+FN}$. Here, True Positive (TP) is the count of blemished onions in the bin, False Positive (FP) is the count of good onions in bin mistaken as being blemished, True Negative (TN) is how many good onions remain on table, and False Negative (FN) is how many blemished onions remain on table mistaken as being good. From Fig. 1, we note that MMAP-BIRL exhibits a much better precision and recall compared to the previous HiddenDataEM method .

In summary, this use of MMAP in the context of IRL under uncertainty yields a new technique that significantly improves on the previous method in both the accuracy of the learned behavior and run time. We establish its usefulness toward robot learning on a use-inspired task in a complex environment.

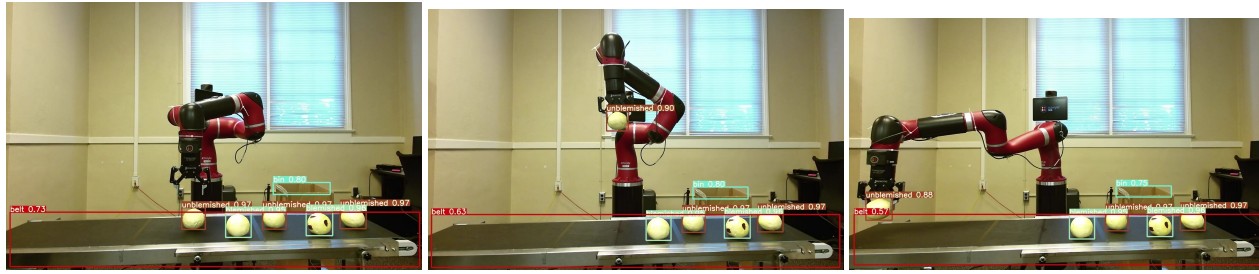

Figure 3: Snapshots of Sawyer sorting through the onions with bounding boxes detected in real time by YOLO v5. The run times were measured on the same computing platform as before.

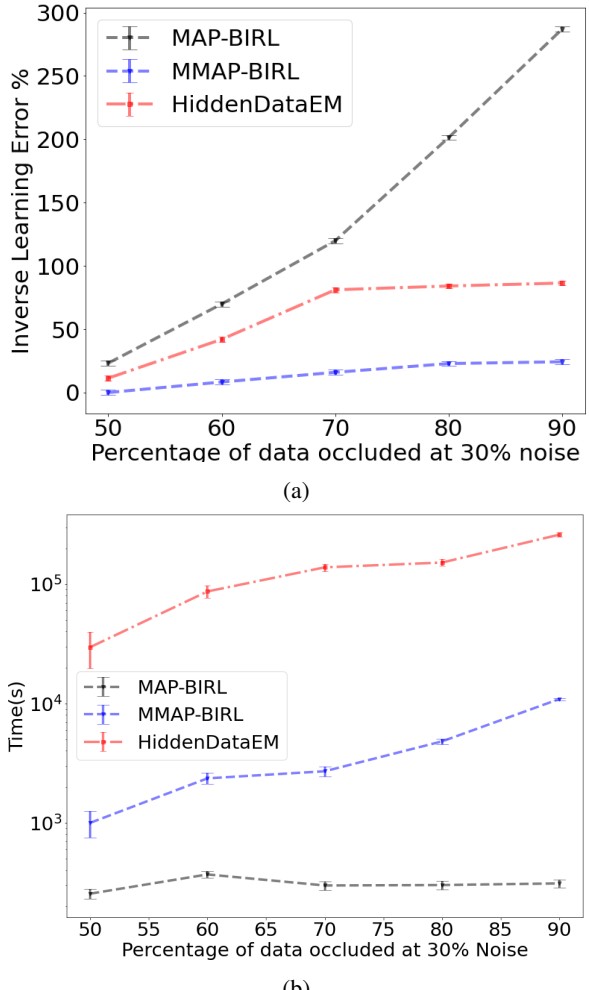

(a)

(b)

Figure 4: (a) MMAP-BIRL exhibits much better ILE performance with varying occlusion % at the estimated 30%-noise level. (b) MMAP-BIRL scales to this larger domain much better than the previous method with increasing occlusions and the same level of noise.

## 5 RELATED WORK

One of the first approaches to consider observer noise [Shahryari and Doshi, 2017] expands the well-known maximum entropy IRL [Ziebart et al., 2008] to maximize the entropy of the joint distribution of the hidden state-action trajectories and observation sequences. A Lagrangian relaxation of this non-linear program yields gradients that can be used in the optimization. However, the gradients are hard to compute making the approach computationally unwieldy and challenging to scale. On the other hand, techniques like Robust-BIRL [Zheng et al., 2014], D-REX [Brown et al., 2020], and the more recent SSRR [Chen et al., 2020] target noisy trajectories due to the expert's failures during task performance. The former is based on the premise that noisy execution may cause the expert to sometimes follow off-policy actions. A latent variable characterizing the reliability of the action is introduced and an expectation-maximization schema in the framework of BIRL manages this noise. The latter technique (D-REX) solves the problem of automatically ranking demonstrated trajectories based on Luce-Shepard rule while SSRR uses Adversarial IRL and assumes that the demonstrator is suboptimal and that pairwise preferences over trajectories are additionally needed for IRL. However, none of these methods introduce an observation model or account for partially occluded trajectories. As the expert in our setting fully and perfectly observes its state while the learner experiences noise due to imperfect sensors, IRL methods that model the expert as a partially observable MDP (POMDP) [Choi and Kim, 2011b] are not relevant.

Over the past few years, there has been a steady stream of methods for inverse learning in the context of occlusion. Beginning with a method that ignores the occlusions (uses just the available data) for maximum entropy IRL [Bogert and Doshi, 2015], to the HiddenDataEM, which inferred the hidden variables in actions using the expectation-maximization schema [Bogert et al., 2016], followed by ways of improving the computational tractability of the approach Bogert and Doshi [2017]. As shown, MMAP-BIRL is a significant improvement over HiddenDataEM. Mai et al. [2019] shows that forward solving the expert's MDP using a system of linear equations also allows for inferring the missing portions of the input data. But, it requires the underlying Markov chain to be non cyclic – an assumption difficult to satisfy in practice.

# 6 CONCLUSION

Motivated by the problem of learning from observing a sorting task on a line, we presented a method to generalize and improve MAP-BIRL to model and reason with both perception noise and unavoidable occlusions of portions of the data. In doing so, we show a new application of MMAP to a domain where the MAP variables are continuous, and developed a gradient-based approach to solve the MMAP inference problem. Results show that MMAP-BIRL significantly improves over the previous maximum-entropy based method for IRL under occlusions and could pave the way for facilitating future cobot deployment on factory floors. The observation model of the camera used in the cobot domain is obtained by empirically estimating the probability of the object classification network YOLO v5 misclassifying onions. Therefore the assumption that the camera model is available is fairly realistic. Based on these results, a promising avenue of future work is to explore multi-agent demonstrations such as multiple humans collaborating on the line task, observe and learn from this collaboration with the ultimate aim of adding a cobot that brings value to the collaboration.

## Acknowledgements

We thank Shibo Li for developing an initial version of the MMAP framework, Kenneth Bogert for providing the HiddenDataEM baseline and assistance with the experimentation, and Senthamil Aruvi for assistance with ROS/Gazebo simulation models. Our work was enabled in part by NSF grant #IIS-1830421 and a Phase 1 grant from the GA Research Alliance to PD.

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
