# OpenReview forum: "Marginal MAP Estimation for Inverse RL under Occlusion with Observer Noise"
_auai.org/UAI/2022/Conference — UAI 2022 Poster_

### Official Review · Reviewer_c25Z · 2022-04-07

**Q2(1) Originality/Novelty:** 2
**Q2(2) Significance/Impact:** 3
**Q2(3) Correctness/Technical Quality:** 3
**Q2(6) Clarity Of Writing:** 3
**Q6 Overall Score:** 6
**Q8 Confidence In Your Score:** 4

**Q1 Summary And Contributions:**

This paper focuses on the problem of inverse reinforcement learning. It is motivated by partial observations are occluded and noisy perception in real-world scenes observations are unavoidable in real-world scenes while existed works cannot well handle these situations. This paper proposes the MMAP-BIRL method to improve IRL which allows learning from both occlusions and noisy perception.

**Q2 Assessment Of The Paper:**

More detailed information regarding each of these aspects is given below:

**Q2(4) Quality Of Experiments (Optional):**

2: Fair: The experimental evaluation is weak: important baselines are missing, or the results do not adequately support the main claims.

**Q2(5) Reproducibility:**

3: Good: Key resources (e.g., proofs, code, data) are available and key details (e.g., proofs, experimental setup) are sufficiently well-described for competent researchers to confidently reproduce the main results.

**Q3 Main Strengths:**

- This paper proposes the MMAP-BIRL method to improve IRL in a more realistic and challenging scenario whose motivation is clear and sufficient.
- This paper provides a relatively clear description of the proposed method. The effect of each part of the method is also clarified.
- This paper is well organized and no obvious syntax errors are found.

**Q4 Main Weakness:**

- The figure and font are not well self-restrained. For example, in Figure 4, the indices are not aligned and the size of subfigures are not consistent.
- The experiment evaluations are not adequate to verify the effectiveness of the proposed methods. More baselines or more evaluation matrices need to be considered for comparison.
- The proposed method relies on somewhat strong assumptions.

**Q5 Detailed Comments To The Authors:**

This paper is well-motivated and the description of the proposed method is clear. However, there are some details worth noting.

- The typesetting of the paper could be improved.
- Some figures are not well self-restrained. For example, the color of each setting is not easy to distinguish which reduce the readability of this paper.
- I have some questions/concerns for this paper:
(1) The rationality for using a Gaussian distribution as a prior could be further explained.
(2) How do you select the hyperparameters (described at the beginning of Section 4)?


**Q7 Justification For Your Score:**

The motivation of this paper is clear and the logic is well organized which is what I weigh most. Moreover, they improve the MMAP model based on MAP and develop a gradient-based method for the MMAP inference problem which makes this paper contributes to this community to some extent. But the experiment evaluations are not adequate to support their claims. The details of this paper also need to be improved.

**Q9 Complying With Reviewing Instructions:**

1: Yes.

---

### Official Review · Reviewer_kWBV · 2022-04-11

**Q2(1) Originality/Novelty:** 2
**Q2(2) Significance/Impact:** 2
**Q2(3) Correctness/Technical Quality:** 4
**Q2(6) Clarity Of Writing:** 4
**Q6 Overall Score:** 6
**Q8 Confidence In Your Score:** 4

**Q1 Summary And Contributions:**

Bayesian formulation of inversement reinforcement learning that allows for state occlusion and lerner's imperfect sensing; gradient-based solution to learn reward function. Experiments with a toyset example a realistic robotic task show the proposed method's advantage over competitors.

**Q2 Assessment Of The Paper:**

More detailed information regarding each of these aspects is given below:

**Q2(4) Quality Of Experiments (Optional):**

3: Good: The experimental evaluation is adequate, and the results convincingly support the main claims.

**Q2(5) Reproducibility:**

4: Excellent: Key resources (e.g., proofs, code, data) are available and key details (e.g., proof sketches, experimental setup) are comprehensively described for competent researchers to confidently and easily reproduce the main results.

**Q3 Main Strengths:**

- Relevant and scientifically challenging problem
- Simple, principled and potentially effective solution
- Clear writing

**Q4 Main Weakness:**

- Proposed approach is straightforward (limited impact)
- Requires handcrafted feature functions
- No discussion on how to extend approach to continuous state and actions
- Experiments consider a single use case and do not test

**Q5 Detailed Comments To The Authors:**

The whole discussion assumes discrete states and actions, however that assumption is not explicitly stated in sections 1 and 2. The discussion in Section 2 and 3 could be summarized and better illustrated with a plate model describing the statistical assumptions of the model, so that most equation would follow directly and could be omitted. The gained space could then be used to elaborate on the experimental results with the robot arm.

There is a hidden assumption that the missing data (occlusion) are missing at random (MAR). For example, occlusion could be associated with a state (e.g. when the human demonstrator behaves in some particular way that obscures the learner's camera). Authors should briefly discuss why such an assumption is reasonable (or consider MAR as an approximation/simplificaiton).

How are the state transition function in the robotic experiment specified/learned?

**Q7 Justification For Your Score:**

The paper pushes the state-of-the-art in inverse reinforcement learning from expert's demonstrations with discrete state-action spaces and known transition function. The proposed method relaxes some of the previous work's limitations (complete observation and perfect sensing) and thus steps towards more realistic settings. The proposed method is a straightforward application of Bayesian inference to the problem, with gradient-based optimization. Experiments show promising results.

**Q9 Complying With Reviewing Instructions:**

1: Yes.

---

### Official Review · Reviewer_cSDR · 2022-04-12

**Q2(1) Originality/Novelty:** 3
**Q2(2) Significance/Impact:** 3
**Q2(3) Correctness/Technical Quality:** 3
**Q2(6) Clarity Of Writing:** 3
**Q6 Overall Score:** 6
**Q8 Confidence In Your Score:** 3

**Q1 Summary And Contributions:**

This is a nice application of MMAP (marginal MAP) to inverse reinforcement learning.
The IRL task (learning the reward function of a MDP from demonstrations) is extended to the case of partially observable (and not only noisy) samples. Experiments on a toy model and on a real application in robotics are promising. The proposed technique is basically a MMAP procedure for continuous "MAR" variables based on gradient ascent.

**Q10 Ethical Concerns (Optional):**

Nothing specific to consider.

**Q2 Assessment Of The Paper:**

More detailed information regarding each of these aspects is given below:

**Q2(4) Quality Of Experiments (Optional):**

3: Good: The experimental evaluation is adequate, and the results convincingly support the main claims.

**Q2(5) Reproducibility:**

3: Good: Key resources (e.g., proofs, code, data) are available and key details (e.g., proofs, experimental setup) are sufficiently well-described for competent researchers to confidently reproduce the main results.

**Q3 Main Strengths:**

The method proposed in this paper extends the class of setups to be used for (Bayesian) inverse RL.
The new setups are of high interest for applications in robotics.
The experiments show the advantages of the new approach (wrt existing methods)

**Q4 Main Weakness:**

The baseline method is designed for a less general setup and, as a matter of fact, the comparison is not fair. Yet, as the authors claim, the method proposed here extend the scope of BIRL so such improved performance is expected.

**Q5 Detailed Comments To The Authors:**

My overall opinion about the paper is positive. As a non-specialist of IRL I see the advantages related to mapping the inference to a MMAP instead of a standard MAP.

Regarding the advantages to not ignoring the "occluded" data, I understand the rationale of this point, but an empirical analysis would make this claim stronger. I also miss a dedicated analysis of the complexity of the overall procedure (and in the empirical part discussing the execution times would be also valuable).

Finally, I could imagine situations where the variables related to the occlusion are discrete. In would be interesting to discuss how to adapt the proposed procedure to such alternative setup.

**Q7 Justification For Your Score:**

I am not a specialist, but I see a clear improvement of the SOTA about (B)IRL provided by this paper. This also establishes a nice link between (MMAP) inference and RL and this sounds very much in the spirit of UAI.

**Q9 Complying With Reviewing Instructions:**

1: Yes.

---

### Official Review · Reviewer_qKfv · 2022-04-13

**Q2(1) Originality/Novelty:** 3
**Q2(2) Significance/Impact:** 3
**Q2(3) Correctness/Technical Quality:** 3
**Q2(6) Clarity Of Writing:** 4
**Q6 Overall Score:** 7
**Q8 Confidence In Your Score:** 3

**Q1 Summary And Contributions:**

This paper presents a generalised approach for the maximum-a-posteriori inverse RL algorithm that handles the noisy information collected while observing and learning with an expert. The proposal marginalises and estimates occluded portions of information using the Bayesian version of MAP-IRL. The marginalised MAP-Bayesian IRL (MMAP-BIRL) were tested on two different domains and against 2 main baselines (MAP-BIRL and the HiddenDataEM).

**Q10 Ethical Concerns (Optional):**

No.

**Q2 Assessment Of The Paper:**

More detailed information regarding each of these aspects is given below:

**Q2(4) Quality Of Experiments (Optional):**

3: Good: The experimental evaluation is adequate, and the results convincingly support the main claims.

**Q2(5) Reproducibility:**

3: Good: Key resources (e.g., proofs, code, data) are available and key details (e.g., proofs, experimental setup) are sufficiently well-described for competent researchers to confidently reproduce the main results.

**Q3 Main Strengths:**

Novelty: The paper presents a novel algorithm that composes well-known approaches into a fine and robust solution.

Impact: The application of the marginal approximation through the Bayesian framework will likely influence the AI community as a possible solution while handling partially observable domains.

Technical Quality: The paper presents an excellent background that justifies and supports the solution presentation. I have not found any issues, but I have not carefully checked the mathematical details.

Quality Of Experiments: The experiments are adequate and the results are enough to support the authors' contribution.

Reproducibility: This work delivers most of the details necessary to reproduce the experiments. I think there is only one piece of information missing: the number of experiments performed.

Writing: The paper is well-organised and clearly written, presenting a good motivation, detailed methodology and clear discussion about the results and development.

**Q4 Main Weakness:**

Novelty: Even considering the solution non-trivial and well-made, I would say it follows the current community ideas to advance the state-of-the-art solutions. This point doesn't disqualify the work.

Impact: Perhaps the paper would have a bigger impact if more complex scenarios were included in the experiments. I think future works, based on these current results, could have an even higher impact.

Quality Of Experiments: I missed the presentation of more results in similar and different scenarios. Even the performance of an ablation study would be appreciated and interesting to see in this work. Presenting more baselines would also corroborate a better evaluation of the current results.

Reproducibility: I missed the number of experiments and the information about which test or method the authors used to calculate the confidence/deviation of their results.

The source code is not included in the supplementary material, but the authors promise to make it available after acceptance.

**Q5 Detailed Comments To The Authors:**

I really appreciate the reading of this work and think it presented a great solution for the community with good results that supports the authors' claim. I kindly suggested improvements above, however it can be addressed in futures works.
I only found two minors while reading:

(1) in the introduction, the emphasis (italics) on the text is somewhat long:

"As such, the learner’s perception of the task performance may be both incomplete and imperfect. This is different from a typical POMDP setup [Choi and Kim, 2011b] as in our case the expert can perfectly observe the environment and the noisy observations are due to the learner’s imperfect sensors."

I think I would highlight only the citation from Choi and Kim's work or the phrase about this work. In the current text, I missed the detail that I should maintain my focus.

(2) in Section 4.2:

"The value ‘hover location’ is a region of space just on top of the conveyor and ’in bin’ indicates just inside a bin that holds (...)"

The apostrophe should be replaced by quotation marks. Moreover, I recommend using ``in bin'' instead of ’in bin’ to guarantee that both quotation marks will be correctly sided.

Additionally, I would like to see an extensive and deeper discussion about the results. I think Fig. 4 could be smaller to save space.

**Q7 Justification For Your Score:**

The paper presents a fine and robust solution for the target domain using relevant state-of-the-art methods to compound the proposed algorithm. The results well-support the authors' claims and the paper is well-organised/clearly written.

**Q9 Complying With Reviewing Instructions:**

1: Yes.

---

### Decision · Program_Chairs · 2022-05-15

**Decision:**

Accept (Poster)

**Comment:**

Meta Review: The paper presents an interesting use of marginal MAP to inverse reinforcement learning extended to the case of partially observable demonstrations. All reviewers agree that the paper has merit as it presents a sound solution to the general and difficult problem of IRL in a partially observable environment. The reviewers also recommended a number of improvements, in particular in the experimental section, which I strongly recommend the authors to address to improve the current version.